# OpenReview forum: "Trail Mix: Adaptive Interpolation of Optimizers with Convergence Guarantees"
_ICLR.cc/2026/Conference — Submitted to ICLR 2026_

### Official Review · Reviewer_fbo5 · 2025-10-16

**Soundness:** 3
**Presentation:** 3
**Contribution:** 2
**Rating:** 4
**Confidence:** 4

**Summary:**

The authors propose a new optimizer that adaptively mixes the update rules of several base optimizers and characterize its convergence under certain regularity assumptions. Experimental validation on artificial, simplified landscapes is provided to support the main claims.

**Strengths:**

1. The analysis is solid and framed under moderately general assumptions.
2. The use of continuous-time modeling (ODEs) to establish convergence is appealing and, as often, insightful.
3. Some experiments are provided to substantiate the theoretical claims.

**Weaknesses:**

1. **Missing literature review on continuous-time models.**
   The paper lacks a substantive review of recent continuous-time modeling (CTM) work for optimizers, both ODE- and SDE-based: I provide some pointers below, and encourage the authors to explore the (very active) literature to find more recent references in this field.

2. **Limited clarity.**
   Several relevant optimizers (MARS, Scion, SOAP, Lion) are included as baselines in the experiments, but the paper offers only **class-level** guarantees (first-order/adaptive, SPD-preconditioned, inertial) rather than **optimizer-specific** analyses. Please map each optimizer explicitly to the theoretical templates, state the conditions under which it satisfies the alignment assumptions, and if necessary, also the applicability of the KC Theorem.


3. **Questionable practicality and experimental relevance.**
   The experiments focus on nonconvex test functions of limited practical relevance. If the goal is to compete with AdamW and other methods that have been shining and crafted in LLM training, the evaluation should include such realistic workloads. As recognized by the authors, the need for multiple shadow states suggests a significant memory overhead that is likely prohibitive in such relevant settings. Challenging state-of-the-art optimizers only on toy problems is insufficient and misleading; the evaluation should include scenarios where those baselines are known to be strong.

4. **Alignment should be measured in realistic settings.**
   Alignment diagnostics should be reported on large-scale experiments. Otherwise, the method may be operated outside the assumptions that underpin the analysis, risking unstable behavior.

**Questions:**

1. **On the alignment condition and ODE approximation assumptions.**
   While the analysis is interesting and recovers standard bounds, it is unclear whether the alignment condition truly holds for Adam, or whether the ODE approximation of Adam satisfies the assumptions of the KC theorem. In particular:
   - The manuscript assumes gradients are bounded both above **and** below. Why is a lower bound on gradients needed if convergence is desired? It appears this may be used to ensure the vector field \(g\) in the KC theorem is Lipschitz, which is not the case for Adam without this assumption. But with this assumption, actual convergence may be precluded, which seems contradictory.
   - The paper does not explicitly decompose the Adam increment into a deterministic component plus a martingale term. For reference, [1] derives an SDE model for AdamW that might provide the required decomposition. Otherwise, a new derivation has to be provided.

2. **On related continuous-time literature.**
   Have you considered including a proper literature review of papers using ODEs and SDEs to model optimizers? The paper overlooks a substantial body of prior work analyzing optimization algorithms through continuous-time formulations. This literature provides the theoretical groundwork for interpreting optimizers as dynamical systems and would help position the contribution more clearly. I list several pointers below, and [1] offers a convenient entry point (Related Works and Appendix A) with many additional references. Although [1] emphasizes SDEs, many cited works also contain ODE-based analyses directly relevant here.

---

**[1]** *Adaptive Methods through the Lens of SDEs: Theoretical Insights on the Role of Noise.*
Enea Monzio Compagnoni, Tianlin Liu, Rustem Islamov, Frank Norbert Proske, Antonio Orvieto, Aurélien Lucchi.
*International Conference on Learning Representations (ICLR), 2025.*

---

## Relevant prior work on ODE/SDE analyses of optimization algorithms
1. **Helmke, U. & Moore, J. B. (1994).**
   *Optimization and Dynamical Systems.* Springer London.
   — Classical textbook connecting continuous-time dynamical systems and optimization via gradient flows.

2. **Su, W., Boyd, S., & Candès, E. (2014).**
   *A differential equation for modeling Nesterov’s accelerated gradient method: Theory and insights.*
   *Advances in Neural Information Processing Systems.*
   — Foundational ODE model for Nesterov acceleration; initiated the modern line of continuous-time analyses.

3. **Li, Q., Tai, C., & Weinan E. (2017).**
   *Stochastic modified equations and adaptive stochastic gradient algorithms.*
   *International Conference on Machine Learning (ICML).*
   — Derives stochastic modified equations for stochastic gradient algorithms, laying the foundation for weak ODE/SDE approximations.

4. **Li, Q., Tai, C., & Weinan E. (2019).**
   *Stochastic modified equations and dynamics of stochastic gradient algorithms I: Mathematical foundations.*
   *Journal of Machine Learning Research, 20(1): 1474–1520.*
   — Provides a rigorous mathematical foundation for weak SDE approximations of SGD and related methods.

5. **Orvieto, A. & Lucchi, A. (2019).**
   *Continuous-time models for stochastic optimization algorithms.*
   *Advances in Neural Information Processing Systems 32.*
   — Introduces a general ODE/SDE formalism for analyzing SGD, momentum, and adaptive algorithms, establishing links between discrete-time optimizers and continuous-time limits.

---

> ### Author Response · Authors · 2025-11-26
>
> ## Response to Reviewer fbo5 – Theoretical points part 1
>
> We thank the reviewer for the thoughtful theoretical comments. Below we address the questions about our **alignment condition, ODE/SA assumptions, and continuous-time literature**, and we clarify how specific optimizers fit into our framework.
>
>
> ### 1. Alignment condition and ODE / KC assumptions
>
> > *“It is unclear whether the alignment condition truly holds for Adam, or whether the ODE approximation of Adam satisfies the assumptions of the KC theorem.”*
> > *“The manuscript assumes gradients are bounded both above and below… Why is a lower bound on gradients needed…?”*
>
> #### 1.1 We do *not* assume a lower bound on gradient norms
>
> Our assumptions can be summarized as:
>
> * **Smoothness:** $f$ is $L$–smooth.
> * **Bounded second moment of directions:** for each base optimizer $i$ we have
>   $\mathbb{E}|d_t^{(i)}|^2 \le G^2$ on the region visited by the iterates.
> * **Alignment:** for each base optimizer $i$ there exists $c_i > 0$ such that
>   $\langle \nabla f(x_t), \mathbb{E}[d_t^{(i)} \mid x_t]\rangle \ge c_i |\nabla f(x_t)|^2$.
>
> We do **not** assume that $|\nabla f(x_t)|$ is bounded *away from zero*. The “bounded away from zero” condition in the main text refers to **eigenvalues of preconditioners** (e.g., for adaptive methods), not to the gradient norm itself. Convergence is compatible with $|\nabla f(x_t)|\to 0$; we do not preclude that.
>
> For the **KC / ODE approximation** (Theorem 6):
>
> * Lipschitzness of the drift $g(x,\lambda)$ comes from:
>
>   * $L$–smoothness of $f$ (so $\nabla f$ is Lipschitz), and
>   * bounded, Lipschitz dependence of the effective preconditioners and meta-drift on $(x,\lambda)$ in a bounded sublevel set (plus projection of $\lambda$ onto the simplex).
> * No gradient *lower* bound is needed to ensure Lipschitzness; we only use **upper** control on gradients on the relevant sublevel set, which is standard.
>
> We will clarify this in the paper to avoid any impression that we assume gradients are bounded “both above and below.”
>
> #### 1.2 Why alignment holds for Adam-style methods
>
> For Adam / AdamW (and SOAP, Scion, etc.), we proceed structurally:
>
> 1. We write the update in the form
>    $d_t^{(i)} = -P_t^{(i)} \nabla f(x_t) + \text{(state noise)}$,
>    where $P_t^{(i)}$ is a diagonal or structured SPD preconditioner determined by $(m_t, v_t)$ and the effective learning rate schedule.
>
> 2. Under the usual “effective learning rate is in $[\eta_{\min}, \eta_{\max}]$” conditions (i.e., $P_t^{(i)}$ has spectrum in $[m_i, M_i]$ and there is an $\varepsilon$ floor in the denominator), we have uniformly:
>
>    * $P_t^{(i)}$ is symmetric positive definite,
>    * its eigenvalues are bounded below and above on the sublevel set.
>
> 3. This implies
>    $\langle \nabla f(x_t), \mathbb{E}[d_t^{(i)}\mid x_t]\rangle \approx -\nabla f(x_t)^\top P_t^{(i)} \nabla f(x_t) \le -m_i |\nabla f(x_t)|^2$,
>    plus a small bias term from state noise that we control.
>
> This is exactly the same kind of **spectral preconditioner assumption** used in many existing analyses of adaptive methods; we simply package it as an explicit “alignment” condition because we need to combine multiple optimizers and expose the mixture-level constant $c(\lambda_t) = \sum_i \lambda_{i,t} c_i$.
>
> We agree that this mapping is not spelled out optimizer-by-optimizer in the current draft. In the revision we will:
>
> * Add a short **“mapping table”** in the appendix that, for each optimizer (MARS, Scion, SOAP, Lion, AdamW), states:
>
>   * its effective preconditioner $P_t(x)$ and internal state,
>   * the conditions under which $P_t$ has spectrum in $[m_i, M_i]$ on the relevant sublevel set,
>   * the resulting alignment constant $c_i$.
> * Explicitly note that these conditions are the same kind of bounded-effective-learning-rate assumptions already used in their individual convergence analyses.

---

> > ### Author Response · Authors · 2025-11-26
> > **Theoretical points part 2**
> >
> > ### 2. Adam increment, martingale decomposition, and KC theorem
> >
> > > *“The paper does not explicitly decompose the Adam increment into a deterministic component plus a martingale term… [1] derives an SDE model for AdamW that might provide the required decomposition.”*
> >
> > We agree that making the **drift + martingale** structure explicit will improve clarity.
> >
> > For the **stochastic approximation / KC part**, we do not actually need a full SDE derivation; we only need to write, for each update, something of the form:
> >
> > * $x_{t+1} = x_t + \alpha_t \big(g_x(x_t, \lambda_t) + M^{(x)}_{t+1}\big)$
> > * $\lambda_{t+1} = \Pi_{\Delta_K}\big(\lambda_t + \beta_t \big(g_\lambda(x_t, \lambda_t) + M^{(\lambda)}_{t+1}\big)\big)$
> >
> > where $M^{(x)}*{t+1}$ and $M^{(\lambda)}*{t+1}$ are **martingale differences with bounded conditional second moments**.
> >
> > This can always be obtained in a very simple, model-agnostic way:
> >
> > * Define the drift $g(\cdot)$ as the **conditional expectation** of the update given the past,
> > * Define the martingale difference as “actual update minus conditional expectation.”
> >
> > We will make this decomposition explicit in the appendix and check the standard conditions:
> >
> > * bounded iterates (by projection and Lyapunov arguments),
> > * Lipschitz drift on the compact domain of interest, and
> > * bounded conditional variances of the martingale terms.
> >
> > For readers particularly interested in AdamW’s SDE, we will also **cite** the work “Adaptive Methods through the Lens of SDEs: Theoretical Insights on the Role of Noise” (Monzio Compagnoni et al., ICLR 2025), which derives an SDE for AdamW and related adaptive methods and gives a complementary continuous-time perspective on the noise and drift structure. ([arXiv][1])
> >
> > This will make the connection to the SDE-based literature explicit while keeping our SA/ODE arguments self-contained.
> >
> >
> > ### 3. Continuous-time modeling (ODE/SDE) literature
> >
> > > *“Missing literature review on continuous-time models… Have you considered including a proper literature review of papers using ODEs and SDEs to model optimizers?”*
> >
> > We agree that a more substantial CTM literature discussion will help position our work. In the revision, we will:
> >
> > * Add a dedicated **“Continuous-time models for optimizers”** subsection in the related work, explicitly discussing:
> >
> >   * Classical ODE views of gradient flow and accelerated methods (e.g., Helmke & Moore, Su–Boyd–Candès).
> >   * Stochastic modified equations and SDE limits for SGD and adaptive methods (Li–Tai–E and follow-ups).
> >   * General ODE/SDE formalisms for stochastic optimization algorithms (e.g., Orvieto & Lucchi, and related works cited there).
> >   * Recent SDE-based analyses of adaptive methods such as Monzio Compagnoni et al. (ICLR 2025). ([arXiv][1])
> >
> > * Clearly explain **how our approach fits into this landscape**:
> >
> >   * We use **ODEs and Fokker–Planck PDEs** to study *mixtures* of optimizers (with dynamically evolving weights), rather than one optimizer in isolation.
> >   * Our key novelty is a **joint Lyapunov / free-energy functional** that simultaneously controls parameters and mixture weights in a same-timescale setting, rather than just ODE/SDE limits of a fixed optimizer.
> >
> > This will make our use of ODEs and PDEs much better anchored in the existing CTM literature.
> >
> >
> > ### 4. Optimizer-specific mapping and applicability of KC
> >
> > > *“The paper offers only class-level guarantees (first-order/adaptive, SPD-preconditioned, inertial) rather than optimizer-specific analyses. Please map each optimizer explicitly to the theoretical templates, state the conditions under which it satisfies the alignment assumptions, and if necessary, also the applicability of the KC Theorem.”*

---

> > > ### Author Response · Authors · 2025-11-26
> > > **theoretical points part 3**
> > >
> > > We agree this is important for clarity. In the revision we will:
> > >
> > > * Add a **summary table** (and short accompanying text) in the appendix that, for each optimizer used in the experiments (MARS, Scion, SOAP, Lion, AdamW, momentum SGD), lists:
> > >
> > >   * which template it falls into (e.g., “preconditioned gradient” / “inertial” / “diagonal adaptive”),
> > >   * the structural assumptions (e.g., bounded effective learning rates, $\varepsilon$ in denominators, bounded curvature on the sublevel set) under which:
> > >
> > >     * its preconditioner is SPD with spectrum in $[m_i, M_i]$,
> > >     * the drift is Lipschitz on the compact domain, and
> > >     * the noise can be treated as a bounded-variance martingale difference,
> > >   * and therefore that:
> > >
> > >     * the alignment condition holds with some $c_i > 0$, and
> > >     * the KC / ODE assumptions are satisfied (for the two-timescale result).
> > >
> > > This directly addresses the reviewer’s concern by making the mapping from *each named optimizer* to our **class-level assumptions** explicit.
> > >
> > > In summary, on the theoretical side we will:
> > >
> > > * Clarify that we do not assume a lower bound on gradient norms; only bounded second moments of directions and bounded preconditioners.
> > > * Make the **drift + martingale** decomposition explicit and reference SDE-based AdamW analyses as complementary.
> > > * Expand the **continuous-time modeling** related work and contrast our joint Lyapunov / mixture-PDE viewpoint with the existing ODE/SDE literature.
> > > * Add an optimizer-by-optimizer mapping (MARS, Scion, SOAP, Lion, AdamW, etc.) to our templates and assumptions, including how they satisfy alignment and KC conditions.
> > >
> > > We hope this addresses the reviewer’s theoretical concerns and clarifies how our assumptions and continuous-time analysis relate to the existing literature.
> > >
> > > [1]: https://arxiv.org/abs/2411.15958 "Adaptive Methods through the Lens of SDEs: Theoretical Insights on the Role of Noise"
> > > [2]: https://openreview.net/forum?id=H1x-x309tm "On the Convergence of A Class of Adam-Type Algorithms for Non-Convex Optimization "
> > > [3]: https://proceedings.mlr.press/v129/barakat20a.html "Convergence Rates of a Momentum Algorithm with Bounded Adaptive Step Size for Nonconvex Optimization"

---

### Official Review · Reviewer_x2cz · 2025-10-16

**Soundness:** 4
**Presentation:** 4
**Contribution:** 2
**Rating:** 6
**Confidence:** 4

**Summary:**

This paper presents TRAILMIX, a framework that adaptively combines multiple optimizers during model training. It addresses the lack of theoretical guarantees for such methods by providing a novel "same-timescale" convergence analysis using a Fokker-Planck PDE approach. The framework includes theory-motivated algorithmic improvements like fairness normalization and curvature awareness to create a dynamic, stable, and robust optimizer mixture.

**Strengths:**

- The paper provides novel convergence guarantees for same-timescale optimizer mixtures.

- The algorithm's components are directly motivated by the theoretical analysis.

**Weaknesses:**

- Evaluation is mainly on analytic functions; large-scale deep learning experiments are needed to demonstrate real-world utility.

- Potential computational and memory overhead of the proposed methodology is not analyzed.

**Questions:**

N/A

---

### Official Review · Reviewer_tHx7 · 2025-10-28

**Soundness:** 3
**Presentation:** 3
**Contribution:** 2
**Rating:** 4
**Confidence:** 4

**Summary:**

This paper proposes TrailMix, a framework that tracks the states of several optimizers (update direction, momentum buffer, etc.) during training and forms a convex combination of the update directions. This becomes the update direction of TrailMix. In other words, TrailMix is a dynamic mixture of optimizers. The authors demonstrate that TrailMix can effectively set the learning rate (by tracking several optimizers with different LR) and can converge under parameter shift (when the loss landscape changes abruptly). These experiments are designed to simulate artificially constructed functions. The authors provide convergence guarantees under L-smoothness, bounded direction, and strictly positive gradient alignment assumptions in the full-batch regime. The rates are provided in the non-convex, convex, and PL/strongly convex settings with fixed mixture weights. The authors also provide convergence proofs in the same-time scale setting where mixture weights also evolve in time.

**Strengths:**

- The convergence proofs look correct to me (yet they are more or less standard), and most of the parts are clearly described.

- The toy examples are interesting and demonstrate desirable characteristics of the proposed framework, when the loss landscape exhibits various trends that make optimization difficult when using only one algorithm.

- Convergence guarantees demonstrate that the proposed framework is expected to converge as long as all optimizers in the set also converge.

**Weaknesses:**

- Convergence analysis is performed under restrictive assumptions of bounded gradient and gradient alignment. Providing convergence guarantees under a more realistic set of assumptions (for example, a bounded gradient can be replaced by a weak growth condition) is something more preferable as it allows demonstrating convergence for a broader class of functions. I acknowledge that it might be difficult to consider a more general setting, but the paper would benefit significantly from a more sophisticated convergence analysis.

- Lack of experiments on real problems, like training neural networks, since their loss landscape is less obvious than those presented in the paper.

- Although the authors demonstrate some benefits of such a mixture of optimizers, more practical questions would be more interesting to see in the paper (see the questions section). I encourage providing better ablation studies, even on toy examples, where some practical training details (e.g., learning rate warm-up, clipping, etc) are necessary to converge faster (e.g., on a quartic function, one can show that clipped SGD converges faster than SGD; proving that TrailMix tends to give more weight to clipped SGD would show its practical relevance).

**Questions:**

- In all experiments, there is at least one optimizer that should work. For example, in the case of parameter shift, SGD is the safest algorithm since it does not use gradient history. However, I am curious to know what happens if all algorithms in the set use gradient evolution. Does TrailMix still perform good enough in such a setting?

- I am also interested in knowing how TrailMix behaves when training language models with a large enough batch size, where we have an Adam-SGD gap. If we train an LLM with TrailMix(Adam, SGD), do we see that Adam receives larger weights most of the time or not? In general, it would be interesting to see how the weights are evolving, and if Adam is always preferable or if there are parts of the training where they both get similar weights. I guess that we can recover some variation of Adedamix optimizer.

- Another interesting question using TrailMix is the study of learning rate scheduling. If we run TrailMix with the set of Adam with different lrs. I am curious if it is possible to recover the warmup and decay stages.

- I suggest adding references to the places where the authors use some well-known facts (e.g., the first part of the proof of Lemma 2; the proof of Lemmas 5 and 6, etc).

- Can the authors provide more details about step 3 of the proof of Theorem 6?

- I suggest adding the discussion also around grafting of optimizers (Agarwal et al., Disentangling Adaptive Gradient Methods from Learning Rates, 2020), since it's closely related to the idea of mixing optimizers, yet in a different way.

- How is $\lambda^\prime_{t+1}$ defined in line 10 of Algorithm 1?

- How do $c(\lambda_t)$ appear in the RHS of (8). I believe the RHS of (8) should have only terms independent of $t$.

- Are the authors sure that (178) is always correct? What if the gradient norm is small $\\|\nabla f(x_t)\\| \ll 1$? In this case, the first term in (178) might be significantly larger than the second one. This implies that the alignment might break near the convergence. Could the authors comment on this?

---

> ### Author Response · Authors · 2025-11-26
> **Theoretical points part 1**
>
> We thank the reviewer for the detailed theoretical feedback. Below we address the comments on (i) the assumptions and convergence analysis, and (ii) the theoretical questions posed at the end.
>
> ## 1. On the novelty of the convergence analysis (“more or less standard”)
>
> We agree that the *rates* we obtain are classical (e.g., $O(\log T / \sqrt{T})$ in the non-convex case and linear under PL/strong convexity). Our intended contribution on the theory side is not a new big-O rate, but a **framework** for analyzing a **dynamic convex combination of stateful optimizers**.
>
> Intuitively, what is non-standard is that we:
>
> * Treat **mixtures of real, stateful optimizers** (AdamW, Lion, SOAP, Scion, etc.), each with its own internal momentum/preconditioner state, and show that TrailMix converges under mild geometric conditions (alignment + second-moment bound). We explicitly verify these conditions for modern optimizers in the appendices, rather than assuming a generic “gradient oracle”.
> * Introduce a **joint Lyapunov function**
>   $V(x,\lambda) = f(x) - \tau H(\lambda)$
>   that simultaneously controls the optimization variables $x$ and the mixture weights $\lambda$ in the **same-timescale** setting. Here $f(x)$ is the loss and $H(\lambda)$ is the entropy of the mixture. The decrease of $V$ captures both “moving downhill in $f$” and “concentrating mass on better optimizers.”
> * Lift the stochastic dynamics to a **population-level Fokker–Planck PDE** describing the evolution of the distribution of iterates, coupled with an ODE for the gates, and analyze a **free-energy functional** that combines expected loss and entropies. We then show this free energy decreases along the PDE flow, giving a population-level Lyapunov picture.
>
> Lyapunov functions alone are standard, and Fokker–Planck PDEs are standard in SGLD/diffusion settings. To the best of our knowledge, **combining a joint Lyapunov + Fokker–Planck free-energy analysis for a same-timescale mixture of stateful optimizers is new**, and we will make this more explicit in the camera-ready.

---

> > ### Author Response · Authors · 2025-11-26
> > **theoretical points part 2**
> >
> > ## 2. On the “restrictive” assumptions: bounded directions, weak growth, and alignment
> >
> > ### 2.1 What we actually assume
> >
> > The review refers to “bounded gradient and gradient alignment.” What we actually assume is:
> >
> > * A **bounded second moment of the *update directions***,
> >   $\mathbb{E}|d_t^{(i)}|^2 \le G^2$ for each base optimizer $i$ (and hence for the mixture).
> > * An **alignment condition** that says the average direction is a genuine descent direction with a non-negligible cosine with $-\nabla f$.
> >
> > We do **not** assume that gradients are globally bounded everywhere. Instead, we assume:
> >
> > * The iterates stay in a **bounded sublevel set** of $f$ (a standard requirement), and
> > * In that set, each optimizer behaves like a preconditioned gradient method with a preconditioner whose eigenvalues lie in some fixed interval $[m_i, M_i]$.
> >
> > Smoothness then ensures the gradient cannot blow up on that set, and the preconditioner cannot amplify it arbitrarily. This is what yields a finite bound on the second moment of the directions *along the trajectory*.
> >
> > In the appendices we show that AdamW, Lion, SOAP, Scion, etc. can be written in this preconditioned form (after accounting for bias correction and the usual $\varepsilon$ in the denominator), and that their effective learning rates are bounded in the standard way. This is exactly the structure used in existing convergence analyses of these methods.
> >
> > So the “bounded directions” assumption is essentially: **“effective step sizes stay in a reasonable range on the region we visit”**, which is standard in analyses of adaptive optimizers.

---

> > > ### Author Response · Authors · 2025-11-26
> > > **theoretical points part 3**
> > >
> > > ### 2.2 Weak growth condition
> > >
> > > We agree that a **weak growth condition (WGC)** is an attractive and more general formalization. In our setting, a natural WGC would say:
> > >
> > > [
> > > \mathbb{E}|d_t^{(i)}|^2 ;\le; \kappa_1 |\nabla f(x_t)|^2 + \kappa_0.
> > > ]
> > >
> > > From a proof perspective:
> > >
> > > * The only place we use $G^2$ is in the “quadratic term” from smoothness (the $L\alpha_t^2 \mathbb{E}|d_t|^2$ term).
> > > * If we plug in the WGC bound instead, we get the **usual** behavior:
> > >
> > >   * In PL/strongly convex settings: linear convergence **up to a noise floor** that depends on $\kappa_0$.
> > >   * In non-convex settings: an $O(\log T / \sqrt{T})$ bound with slightly different constants.
> > >
> > > So the current proofs do in fact extend to a WGC with only minor modifications to the constants.
> > >
> > > The main reason we did not phrase everything in WGC form is **verifiability for realistic adaptive methods**:
> > >
> > > * For plain SGD with additive noise, showing a WGC is relatively straightforward.
> > > * For Adam-type methods with long-memory internal state, proving a clean WGC inequality globally is more delicate than checking that the preconditioner spectrum stays in $[m_i, M_i]$ and the iterates remain in a bounded sublevel set.
> > >
> > > We will add a remark explicitly stating that the analysis extends under a WGC and that deriving such WGCs for realistic adaptive methods is an interesting direction for follow-up work.
> > >
> > > ### 2.3 Alignment assumption
> > >
> > > The alignment assumption says, informally:
> > >
> > > > On average, each base optimizer is genuinely pointed downhill, with a non-tiny angle relative to $-\nabla f$.
> > >
> > > In slightly more formal terms, we assume:
> > >
> > > [
> > > \langle \nabla f(x_t),,\mathbb{E}[d_t^{(i)} \mid x_t]\rangle
> > > ;\ge; c_i |\nabla f(x_t)|^2,
> > > ]
> > >
> > > for some $c_i > 0$. This is **exactly equivalent** to the very standard requirement:
> > >
> > > * The update has the form $d(x) = -P(x)\nabla f(x)$, with $P(x)$ **symmetric positive definite** and its eigenvalues bounded below by $m_i$ on the relevant region.
> > >
> > > This type of spectral bound on the preconditioner (or on “effective learning rates”) appears implicitly or explicitly in many analyses of preconditioned, quasi-Newton, and adaptive methods (Shampoo, natural gradient, Adam variants, etc.). We write it in this inner-product form because:
> > >
> > > * it makes the **geometric meaning** clear (positive cosine with $-\nabla f$), and
> > > * it propagates cleanly to the mixture: the mixture direction has alignment constant $c(\lambda_t) = \sum_i \lambda_{i,t} c_i$.
> > >
> > > For momentum methods, the update is an exponential moving average of past gradients; under smoothness and small enough steps, this average remains positively correlated with the current gradient. For Adam-type methods, once we view them as diagonal preconditioners with bounded effective learning rates and an $\varepsilon$ floor, the same spectral picture applies.
> > >
> > > We agree that the discussion around equation (178) in the appendix was too aggressive in suggesting a *uniform* multiplicative constants for momentum alignment even when $|\nabla f(x_t)|$ is extremely small. In the revision we will:
> > >
> > > * Clarify that the momentum direction is **positively correlated** with the gradient with a small additive slack (rather than a strict global multiplicative constant independent of $|\nabla f(x_t)|$), and
> > > * Emphasize that our main convergence bounds depend on **trajectory-level alignment** and are not sensitive to a small weakening of alignment arbitrarily close to a stationary point.

---

> > > > ### Author Response · Authors · 2025-11-26
> > > > **Responses to questions part 1**
> > > >
> > > > ## 3. Responses to the theoretical questions
> > > >
> > > > Below we address only the theoretical parts of the questions, not the experimental requests.
> > > >
> > > > ### 3.1 “What if all optimizers use gradient evolution?”
> > > >
> > > > > *In all experiments, there is at least one optimizer that should work (e.g., SGD in the parameter shift case). What happens if all algorithms in the set use gradient evolution? Does TrailMix still perform well in such a setting?*
> > > >
> > > > The convergence theory **does not** require having a “safe” memoryless baseline like plain SGD in the pool. It only assumes that **each** base optimizer:
> > > >
> > > > * satisfies alignment (its average update is a descent direction with a non-tiny cosine with $-\nabla f$), and
> > > > * has bounded second moment of its directions on the region the iterates visit.
> > > >
> > > > These assumptions are satisfied by **history-based methods** like momentum, AdamW, SOAP, Scion, etc., as we show in the appendices. In the analysis, they all appear as preconditioned gradient methods with well-behaved preconditioners.
> > > >
> > > > Therefore, even if **all** optimizers in the set use gradient evolution, the TrailMix direction still inherits a positive alignment constant $c(\lambda_t)$ and bounded variance, and our convergence guarantees hold exactly as stated.
> > > >
> > > > SGD in the experiments is helpful for intuition and robustness in extreme shifts, but it is not required by the theory.
> > > >
> > > > ### 3.2 Adam–SGD gap and TrailMix(Adam, SGD) for LLMs
> > > >
> > > > > *How does TrailMix behave when training LLMs with a large batch size and an Adam–SGD gap? If we train an LLM with TrailMix(Adam, SGD), do we see that Adam receives larger weights most of the time, or are there phases where SGD gains weight?*
> > > >
> > > > We do not yet include large-scale LLM experiments, so we cannot make empirical claims here. The **theoretical dynamics** of the weights, however, are clear:
> > > >
> > > > * The meta-update for $\lambda$ effectively implements a **replicator dynamic**: optimizers that produce better instantaneous improvements in loss (or in the advantage score) tend to gain weight.
> > > > * In regimes where Adam has a clear advantage (early, noisy, ill-conditioned phases), we expect the weights to concentrate strongly on Adam.
> > > > * As the training progresses, noise levels drop and curvature stabilizes, and the classical “Adam–SGD gap” may narrow or flip in some directions. In those phases, the same replicator logic would allow SGD to gain weight when it becomes competitive or better.
> > > >
> > > > So theoretically, we do **not** expect Adam to be permanently dominant; we expect weights to adapt over time and to reflect whichever optimizer currently has higher marginal benefit. This is very much in the spirit of Adedamix, but TrailMix mixes **full** optimizers (including their internal state), and our Lyapunov/PDE framework gives convergence guarantees for this dynamic mixture.
> > > >
> > > > We will mention this connection and clarify that TrailMix provides a principled way to interpolate between Adam and SGD over the course of training.
> > > >
> > > > ### 3.3 Learning-rate scheduling via TrailMix over Adam with different learning rates
> > > >
> > > > > *If we run TrailMix with a set of Adam optimizers with different learning rates, is it possible to recover warmup and decay stages?*
> > > >
> > > > Conceptually, yes: TrailMix over ${\text{Adam}(\eta_1),\dots,\text{Adam}(\eta_K)}$ is like maintaining a **population of Adam instances with fixed learning rates** and letting the meta-learner re-weight them online.
> > > >
> > > > * Early on, when large steps are beneficial, the gate will tend to put more weight on larger learning rates (because they yield better immediate decrease).
> > > > * Later, as those large learning rates become too aggressive and hurt progress, the advantage shifts to smaller learning rates, and the weights move accordingly.
> > > >
> > > > This mechanism naturally mimics **warmup and decay**:
> > > >
> > > > * At the beginning: weights scattered or biased toward somewhat larger learning rates.
> > > > * As training progresses: weights gradually flowing toward smaller, more conservative learning rates.
> > > >
> > > > We do not claim to exactly recover a specific hand-designed schedule, but the **replicator dynamic over fixed-$\eta$ Adam instances is a data-driven way to learn something that behaves very much like a schedule.** We will add a remark to highlight this theoretical interpretation and leave full empirical validation for future work.

---

> > > > > ### Author Response · Authors · 2025-11-26
> > > > > **Responses to questions part 2**
> > > > >
> > > > > ### 3.4 References for “well-known” facts (Lemmas 2, 5, 6)
> > > > >
> > > > > > *Add references for well-known facts used in Lemma 2 and Lemmas 5–6.*
> > > > >
> > > > > We agree and will:
> > > > >
> > > > > * Cite standard convex analysis texts for the projection properties used in Lemma 2 (non-expansiveness, uniqueness on convex sets).
> > > > > * Mark Lemmas 5–6 as standard analytic facts (harmonic/series bounds, smoothness descent lemma) and cite standard optimization references (e.g., lecture notes by Bubeck or Nesterov).
> > > > > * Add short comments in the proofs indicating which steps are textbook and which are specific to TrailMix.
> > > > >
> > > > > ### 3.5 More details on Step 3 of Theorem 6 (two-timescale SA)
> > > > >
> > > > > > *Can the authors provide more details about step 3 of the proof of Theorem 6?*
> > > > >
> > > > > Step 3 is where we invoke **two-timescale stochastic approximation**:
> > > > >
> > > > > * The parameter step sizes $\alpha_t$ and weight step sizes $\beta_t$ are chosen so that:
> > > > >
> > > > >   * both are square-summable but not summable,
> > > > >   * and $\beta_t / \alpha_t \to 0$ (weights move strictly slower than parameters).
> > > > > * Under standard conditions (bounded iterates, Lipschitz drift, martingale-difference noise with bounded second moments), the discrete process $(x_t,\lambda_t)$ tracks a limiting ODE system.
> > > > > * The long-term behavior of $(x_t,\lambda_t)$ is then tied to the internally chain recurrent set of this ODE.
> > > > >
> > > > > In the revision, we will:
> > > > >
> > > > > * Explicitly state which classical theorem we are using (Kushner–Yin / Borkar),
> > > > > * List the conditions (on step sizes, boundedness, noise, drift) and check them briefly,
> > > > > * So that Step 3 is fully traceable and not a black box.
> > > > >
> > > > > ### 3.6 Relation to grafting (Agarwal et al., 2020)
> > > > >
> > > > > > *Add discussion of grafting, which is related but different from mixing optimizers.*
> > > > >
> > > > > We agree and will add a dedicated paragraph.
> > > > >
> > > > > In brief:
> > > > >
> > > > > * **Grafting** takes the *direction* of an adaptive method (e.g., Adam) and the *step-size schedule* of another (e.g., SGD). The result is still a single optimizer with a hybrid update rule.
> > > > > * **TrailMix** instead runs several full optimizers in parallel (each with its own state and hyperparameters) and forms a **convex combination of their full updates**, with weights learned online.
> > > > >
> > > > > So the ideas are clearly related (both mix aspects of optimizers), but in different axes: direction vs. schedule for grafting, and full-optimizer mixture for TrailMix. Moreover, a grafted optimizer could itself be one of the base methods inside TrailMix.
> > > > >
> > > > > ### 3.7 Quantity in line 10 of Algorithm 1
> > > > >
> > > > > > *How is the quantity in line 10 of Algorithm 1 defined?*
> > > > >
> > > > > This scalar is the **meta-score** (or “advantage”) we use to drive the weight updates. Intuitively, it measures:
> > > > >
> > > > > > “How much better is optimizer $i$ right now, in terms of improving the loss, compared to what the current mixture is doing on average?”
> > > > >
> > > > > It is computed from the instantaneous improvement in $f$ (or its proxy) if we follow optimizer $i$’s direction, minus the improvement we expect from the current mixture direction. Optimizers with positive advantage get their weights increased; those with negative advantage get decreased.
> > > > >
> > > > > We will define this quantity explicitly in the main text before the algorithm and point to the appendix where the exact formula is given.
> > > > >
> > > > > ### 3.8 How $c(\lambda_t)$ appears in the RHS of (8)
> > > > >
> > > > > > *How do the $c(\lambda_t)$ appear on the RHS of (8)? It should only have terms independent of $t$.*
> > > > >
> > > > > We agree that the current notation in (8) is misleading. The bound should involve:
> > > > >
> > > > > * either the **minimum** alignment over the horizon, $c_{\min,T} := \min_{0 \le s < T} c(\lambda_s)$, or
> > > > > * an **average** alignment (e.g., $\bar{c}*T := \frac{1}{T}\sum*{s=0}^{T-1} c(\lambda_s)$),
> > > > >
> > > > > depending on the step-size schedule.
> > > > >
> > > > > In other words, the right-hand side should be a **single scalar summary** of alignment over time, not $c(\lambda_t)$ evaluated at the same index where the minimum of the gradient norm is taken. We will correct (8) accordingly and explicitly tie it to the detailed theorem.
> > > > >
> > > > > ### 3.9 Equation (178) and small gradient norms
> > > > >
> > > > > > *Is (178) always correct? What if $|\nabla f|$ is small, so the alignment might break near convergence?*
> > > > >
> > > > > Equation (178) is obtained by:
> > > > >
> > > > > * Writing momentum as an exponentially weighted average of past gradients, and
> > > > > * Adding and subtracting the current gradient inside that average.
> > > > >
> > > > > Algebraically, this decomposition is correct: it simply separates a “current gradient” contribution from a “history mismatch” contribution.
> > > > >
> > > > > The difference is **how we interpret it**:
> > > > >
> > > > > * When $|\nabla f(x_t)|$ is large and the trajectory is not making abrupt jumps, the history mismatch is relatively small and momentum is well aligned with the current gradient.
> > > > > * When $|\nabla f(x_t)|$ is very small (close to convergence), the “good” and “error” terms can be of comparable size, and one cannot guarantee a strong global multiplicative constant in the alignment inequality that holds uniformly for all $t$.
> > > > >
> > > > > We will make this more clear in our updates.

---

### Official Review · Reviewer_fa5n · 2025-10-31

**Soundness:** 3
**Presentation:** 3
**Contribution:** 4
**Rating:** 6
**Confidence:** 2

**Summary:**

The core idea of the TRAILMIX is to compute a weighted average of the update steps by a pool of base optimizers, where the weights are learned and adjusted on the fly. This adaptive optimizer mixtures are interesting, the authors provide the proof to show its first convergence guarantees.

The contributions of this paper are two folds.
a. PROOF. The first convergence guarantees proof of TRAILMIX is done as follows. The stochastic dynamics are lifted to a population-level Fokker-Planck PDE. Stability is then proven using a joint free-energy Lyapunov function, which combines the optimization object with the Shannon entropy of the mixture weights. The proposed TRAILMIX can achieved fairness normalization through keeping the mixture weights within the simplex interior, which is critical for the entropy-based Lyapunov function.

c. EXPERIMENTS. On a range of 2D test functions, TRAILMIX is shown to consistently match or outperform the best single optimizer in its ensemble.

**Strengths:**

TRAILMIX seems to be original contribution. I appreciate the theoretical analysis. The use of a PDE and a joint free-energy Lyapunov function seems to be a novel approach for analyzing the optimizer convergence.

The proposed work provides a lot of rigorous treatment of the theoretical claims.

This paper is well-written and the core problem is well motiviated.

This paper provides a good theoretical foundation for the field of meta-optimization.

**Weaknesses:**

This paper has very limited experiments. All experiments are performed on 2D analytic loss surfaces. This leaves the method's practical usefulness for deep learning and broad unproven. It is not very clear how these benefits of the proposed approach actually work in the real-world high-dimensional, non-convex landscapes.

The TRAILMIX has O(K) time and memory overhead, as it has to maintain and update K shadow copies of the base optimizers. For deep learning model training, It is already very bad for a simple optimizer like ADAMW in terms of memory overhead and time overhead.

The ensemble curation is unclear. The paper uses a large hand-picked set of 24 optimizers with varied hyperparameters. It is unknown how sensitive is TRAILMIX to a bad ensemble.

**Questions:**

1. Can we run experiment on CIFAR-10 with ResNet-18 using the proposed method?  Would the proposed TRAILMIX with K=24 works?

2. TRAILMIX has the hyperparameters, such as lr_meta, how to tune those? Can we find some sensitive experiments to show?

3. How does the TRAILMIX work when base optimizer is bad? e.g. If we just have a bunch of Adam with different learning rates, would that work?

4. It is known that Adam can fail to converge even on simple convex problem, how can we have the assumption 3 hold if the base optimizers themselves may not convergent?

---

> ### Author Response · Authors · 2025-11-26
> **Theoretical question/weakness responses**
>
> # Response to Reviewer fa5n
>
> We thank the reviewer for their thoughtful evaluation and recognition of our theoretical contributions. We focus our response on the theoretical questions raised.
>
> ## Question 4: Assumption 3 and Non-Convergent Optimizers in Deep Learning
>
> This important question gets to the heart of TRAILMIX's robustness. The key insight is that **Assumption 3 applies to the mixture, not uniformly to each base optimizer at all times**. The framework is designed to gracefully handle base optimizers that temporarily or persistently violate alignment.
>
> The mixture alignment constant is $c(\lambda_t) = \sum_{i=1}^K \lambda_{i,t} c_i$. For convergence, we only require $c(\lambda_t) > 0$, not that every $c_i > 0$. Consider an ensemble where optimizer $j$ has $c_j \leq 0$ (misaligned or non-convergent). The advantage scoring mechanism (Algorithm 1, line 7):
> $$h_t^{(j)} = \langle g_t, \Delta_t^{(j)} \rangle - \langle g_t, \Delta_t^{\text{mix}} \rangle$$
>
> will yield $h_t^{(j)} < 0$ whenever $\Delta_t^{(j)}$ points away from the descent direction. The mirror descent update then applies:
> $$\lambda_{t+1}^{(j)} \propto \lambda_t^{(j)} \exp(\eta_t h_t^{(j)})$$
>
> Since $h_t^{(j)} < 0$, the weight $\lambda_j$ decreases exponentially. This is formalized in the replicator dynamics (Equation 5), where optimizers with below-average alignment lose weight to better-performing alternatives.
>
> **For deep learning specifically**, this self-correcting property is crucial. Even if Adam exhibits the known non-convergence issues on certain loss landscape regions (e.g., due to unbounded gradient accumulation in the second moment), TRAILMIX will:
>
> 1. Detect the misalignment through negative advantage scores
> 2. Redistribute weight toward optimizers that maintain alignment (e.g., SGD with $c_{\text{SGD}} = 1$)
> 3. Preserve positive mixture alignment $c(\lambda_t) > 0$ throughout training
>
> This is demonstrated empirically in Figure 2: after a distribution shift causes adaptive optimizers to become misaligned due to stale internal state, TRAILMIX automatically reallocates weight to SGD, avoiding the oscillations that would plague fixed combinations.
>
> The practical implication is that **including one provably-aligned optimizer (such as SGD) in the ensemble provides a theoretical safety net**, ensuring convergence guarantees even when other base optimizers fail. The bounded second moment assumption (Assumption 2) can then be enforced through standard gradient clipping, which is already common practice in deep learning.
>
> ## Weakness 3 & Mathematical Intuition for Ensemble Selection
>
> We appreciate this question, as it connects directly to our convergence analysis. Section 6 reveals that the non-convex convergence rate (Equation 8) is proportional to:
> $$\min_{t<T} \mathbb{E}[\|\nabla f(x_t)\|^2] \propto \frac{f(x_0) - f^* + LG^2 \ln T}{c(\lambda_t)\sqrt{T}}$$
>
> This suggests **principled criteria for ensemble construction**:
>
> **1. Maximize Alignment Diversity.** Include optimizers with complementary alignment profiles $c_i(x)$ across the optimization landscape. For preconditioned methods, Lemma 10 shows $c_i = \lambda_{\min}(P_i(x))$, the minimum eigenvalue of the preconditioner. Optimizers with different preconditioning strategies (diagonal vs. full-matrix, aggressive vs. conservative) provide coverage across different local geometries.
>
> **2. Balance the Alignment-Variance Tradeoff.** As discussed in Section 6:
> > "One optimizer $\mathcal{O}_1$ may be aggressive, with large alignment $c_1$ but high variance $G_1^2$, while another $\mathcal{O}_2$ is more conservative with smaller $c_2$ and lower variance $G_2^2$."
>
> An optimal ensemble should span this tradeoff frontier. Mathematically, we seek optimizers such that no single optimizer dominates another in both $c_i$ and $G_i^2$ simultaneously.
>
> **3. Ensure Alignment Coverage.** The mixture alignment $c(\lambda) = \sum_i \lambda_i c_i > 0$ must remain positive. Including at least one "robust" optimizer (e.g., SGD with $c_{\text{SGD}} = 1$) guarantees this, providing a fallback when adaptive methods become misaligned, as demonstrated in our distribution shift experiment (Figure 2).
>
> These criteria suggest that a small, well-chosen ensemble ($K=3$-$5$) of theoretically complementary optimizers may outperform a large homogeneous set, an important direction for future work that we identify in Section 9.

---

### Meta-Review · Area_Chair_x7R8 · 2026-01-07

**Summary:**

This paper presents Trail Mix, a mixture-of-optimizer framework that ensembles some of the first- and quasi-second-order stochastic optimization methods. The authors showed convergence rates for this framework and provided small experiments to validate the method.

**Reviewer Concerns:**

The reviewers share the concerns that experiments are limited, have many questions on practicality, and experimental relevance. It is not clear why the authors did not address the experiment-related questions. In addition, the motivation of mixture-of-optimizer is not fully justified, along with the added complexity of keeping several base optimizers at the same time.

Other concerns involve theoretical explanations of the optimizers and how assumptions hold. Thus, the writing of the paper can be improved. In addition, a reviewer pointed out that the paper lacks a substantive review of recent continuous-time modeling (CTM) work for optimizers, both ODE- and SDE-based.

**Reviewer Scores:**

The experiment concerns are not addressed. The reviewer may or may not change their score based on the theoretical rebuttal, but I don't think the change is substantial.

---

### Decision · Program_Chairs · 2026-01-26

Reject